# Correlation of Tumor Pathology with Fluorescein Uptake and MRI Contrast-Enhancement in Stereotactic Biopsies

**DOI:** 10.3390/jcm11123330

**Published:** 2022-06-10

**Authors:** Ran Xu, Judith Rösler, Wanda Teich, Josefine Radke, Anton Früh, Lea Scherschinski, Julia Onken, Peter Vajkoczy, Martin Misch, Katharina Faust

**Affiliations:** 1Department of Neurosurgery, Charité—Universitätsmedizin Berlin, Corporate Member of Freie Universität Berlin, and Humboldt-Universität zu Berlin, and Berlin Institute of Health, 13353 Berlin, Germany; judith.roesler@charite.de (J.R.); wanda.teich@charite.de (W.T.); anton.frueh@charite.de (A.F.); lea.scherschinski@charite.de (L.S.); julia.onken@charite.de (J.O.); peter.vajkoczy@charite.de (P.V.); martin.misch@charite.de (M.M.); katharina.faust@charite.de (K.F.); 2BIH Charité (Junior) (Digital) Clinician Scientist Program, Berlin Institute of Health at Charité—Universitätsmedizin Berlin, BIH Biomedical Innovation Academy, Charitéplatz 1, 10117 Berlin, Germany; 3Department of Neuropathology, Charité—Universitätsmedizin Berlin, Corporate Member of Freie Universität Berlin, and Humboldt-Universität zu Berlin, and Berlin Institute of Health, 13353 Berlin, Germany; josefine.radke@charite.de; 4Institute of Pathology, University of Greifswald, 17475 Greifswald, Germany; 5German Cancer Consortium (DKTK), Partner Site Charité Berlin, 10117 Berlin, Germany

**Keywords:** sodium fluorescein, NaFl, brain tumor, stereotactic biopsy, tumor biopsy, fluorescein-guided surgery, fluorescein-guided biopsy

## Abstract

The utilization of fluorescein-guided biopsies has recently been discussed to improve and expedite operative techniques in the detection of tumor-positive tissue, as well as to avoid making sampling errors. In this study, we aimed to report our experience with fluorescein-guided biopsies and elucidate distribution patterns in different histopathological diagnoses in order to develop strategies to increase the efficiency and accuracy of this technique. We report on 45 fluorescence-guided stereotactic biopsies in 44 patients (15 female, 29 male) at our institution from March 2016 to March 2021, including 25 frame-based stereotactic biopsies and 20 frameless image-guided biopsies using VarioGuide^®^. A total number of 347 biopsy samples with a median of 8 samples (range: 4–18) per patient were evaluated for intraoperative fluorescein uptake and correlated to definitive histopathology. The median age at surgery was 63 years (range: 18–87). Of the acquired specimens, 63% were fluorescein positive. Final histopathology included glioblastoma (n = 16), B-cell non-Hodgkin lymphoma (n = 10), astrocytoma, IDH-mutant WHO grade III (n = 6), astrocytoma, IDH-mutant WHO grade II (n = 1), oligodendroglioma, IDH-mutant and 1p/19q-codeleted WHO grade II (n = 2), reactive CNS tissue/inflammation (n = 4), post-transplantation lymphoproliferative disorder (PTLD; n = 2), ependymoma (n = 1), infection (toxoplasmosis; n = 1), multiple sclerosis (n = 1), and metastasis (n = 1). The sensitivity for high-grade gliomas was 85%, and the specificity was 70%. For contrast-enhancing lesions, the specificity of fluorescein was 84%. The number needed to sample for contrast-enhancing lesions was three, and the overall number needed to sample for final histopathological diagnosis was five. Interestingly, in the astrocytoma, IDH-mutant WHO grade III group, 22/46 (48%) demonstrated fluorescein uptake despite no evidence for gadolinium uptake, and 73% of these were tumor-positive. In our patient series, fluorescein-guided stereotactic biopsy increases the likelihood of definitive neuropathological diagnosis, and the number needed to sample can be reduced by 50% in contrast-enhancing lesions.

## 1. Introduction

Fluorescent contrast agents have been increasingly used for vascular and oncological interventions in neurosurgery. 5-aminolevulinic acid (5-ALA) aids in the microscopic resection of malignant glioma by facilitating intraoperative tumor-visualization and improves oncological outcome [1]. Furthermore, fluorescein-guided surgeries increase the extent of resection for high-grade gliomas [2,3,4]. In non-contrast-enhancing pathologies, fluorescein uptake may also help in detecting tumor tissue and areas of higher proliferative activity [5]. Nevertheless, accumulation patterns of different histological entities such as metastases, meningioma, or low-grade glioma remain heterogeneous and less studied [6].

Fluorescein uptake is the result of an extracellular accumulation in, for example, a tumor-induced environment of a damaged blood–brain barrier, and has therefore a high specificity in identifying contrast-enhancing lesions [3]. Due to the extracellular distribution, fluorescein exhibits a relatively low sensitivity for neoplastic cells [4]. Furthermore, normal brain tissue may also exhibit fluorescein uptake, especially in its non-albumin bound conformation [7].

For inoperable lesions, stereotactic biopsy is the procedure of choice to confirm the histopathological diagnosis and enable an adjuvant treatment. The main disadvantages of this intervention encompass sample-size associated bleedings [8] or sampling errors with regard to histological downgrading or erroneous pathological diagnosis [9,10]. Uncertainty or failure in obtaining conclusive diagnostic tissue can result in treatment delay or additional surgical procedures for patients. The additional usage of fluorescein or 5-ALA fluorescence-guidance for frame-based or frameless biopsies has been suggested to overcome these limitations by reducing sample size and ensuring an exact diagnosis in high-grade gliomas [8,11,12,13,14,15]. However, former studies comprise mainly retrospective data collection and small study cohorts. Moreover, uncertainty remains regarding an additional value for non-contrast-enhancing pathologies or tumors other than glioblastomas.

In this study, we aim to report our experience with fluorescein-guided biopsies of contrast- and non-contrast-enhancing lesions, as well as to elucidate distribution patterns in different histopathological diagnoses.

## 2. Materials and Methods

From March 2016 to March 2021, 44 consecutive patients (29 male and 15 female) were prospectively enrolled to undergo fluorescence-guided stereotactic biopsy of an intracranial lesion at our institution. One patient had a second biopsy due to inconclusive pathology (reactive central nervous system tissue). Patients received a dosage of 5 mg/kg Fluorescein Alkon i.v. intraoperatively. Intracranial pathology was diagnosed in all patients using a high-resolution (1.5- or 3-tesla) MRI (magnetic resonance imaging) dataset (T1-MPRAGE) with and without the administration of a gadolinium-based contrast agent, as well as T2-weighted or FLAIR (fluid attenuated inversion recovery) sequences.

Inclusion criteria comprised patients eligible for general anaesthesia, over 18 years of age, and ability to provide written informed consent. Patients were excluded in the case of fluorescein-intolerance, a participation in an experimental adjuvant treatment study, renal failure, bronchial asthma, beta-blocker antihypertensive treatment, cardiac ejection fraction of <40%, and blood pressure instability during anaesthesia.

Patients were eligible for frameless (Brainlab VarioGuide^®^, Brainlab AG, Munich, Germany) or frame-based (Leksell Stereotactic System, Elekta^®^, Stockholm, Sweden) stereotactic biopsy if the lesion appeared inoperable and/or of unknown histology, e.g., suspected radiation necrosis, inflammation, or lymphoma. For deep-seated, small (<10 mm) and/or eloquent lesions, the procedure of choice was frame-based stereotactic biopsy. For larger and more superficial pathologies, the VarioGuide^®^ system was used. Preoperatively, a precise trajectory-planning was carried out, as well as a calculation of the required sample size with an inter-biopsy distance of 0 mm. In case of non-contrast-enhancing lesions, fusion of the T2-weighted and MPRAGE-datasets was performed.

The biopsy technique for each modality is illustrated in Figure 1A,B. Briefly, in frame-based biopsies, a grasping needle of the Riechert–Mundinger system was used, with the first biopsy taken in the most distant part of the tumor and with each consecutive specimen moved proximally through the tumor core until the transition zone was reached. In frame-based biopsies, the Sedan side-cutting needle was typically used. Here, the initial biopsy was taken in the transition zone and then further advanced to the tumor core and up to the tumor margin. Due to these different biopsy needles, specimens were different in size, approximately 2 mm in diameter in frame-based biopsies, as compared to approximately 8 mm length in side-cutting needles using the frameless technique.

The biopsy samples were evaluated intraoperatively for fluorescein uptake under the Pentero 900 microscope with Y560 filter (Carl Zeiss, Meditec^®^, Oberkochen, Germany) (Figure 1C). Each sample was then examined by a consultant neuropathologist for the histopathological diagnosis according to the 2016 WHO classification of tumors of the central nervous system [16], as well as other localization-specific characteristics (Figure 1D), which were then further correlated with fluorescein uptake. A total number of 383 biopsy samples were obtained with a median of 8 specimens (range: 4–18) per patient. Four cases (2 glioblastomas, 1 AA III, and 1 ependymoma) had to be excluded due to the lack of information of the specific fluorescein uptake patterns, leaving a total number of 41 cases and 347 biopsy samples for the final statistical analysis.

The study was approved by the local ethics committee (approval number: EA1/089/21) of the Charité University Hospital and conducted in accordance with the Declaration of Helsinki. All patients provided written informed consent for the off-label use of i.v. fluorescein-guided biopsy.

Values are presented as mean +/− standard deviation (SD), unless otherwise stated. Statistical analysis was performed using GraphPad Prism software (Version 7.0). Elements of Figure 1 were composed using BioRender.com (accessed on 27 April 2022).

## 3. Results

### 3.1. Patient Characteristics

A total number of 383 biopsies were obtained in 45 biopsy cases in our study cohort. The patient characteristics are summarized in Table 1.

The median age at surgery was 63 years (range: 18–87); 15 of the patients were female, and 29 were male. The final histopathology included glioblastoma, IDH-mutant or IDH-wildtype WHO grade IV (GBM; n = 16), B-cell non-Hodgkin lymphoma (n = 10), astrocytoma, IDH-mutant WHO grade III (n = 6), astrocytoma, IDH-mutant WHO grade II (n = 1), oligodendroglioma, IDH-mutant and 1p/19q-codeleted WHO grade II (n = 2), reactive CNS tissue/inflammation (n = 4), post transplantation lymphoproliferative disorder (PTLD; n = 2), ependymoma (n = 1), infection (toxoplasmosis; n = 1), multiple sclerosis (n = 1), and metastasis (n = 1).

### 3.2. Surgical Biopsy and Postoperative Course

All surgical biopsies were performed without intraoperative complications. Three patients (6.7%) showed relevant hemorrhages on postoperative CT with neurological deficits, two of which were managed conservatively and fully recovered and one of which had a prolonged course in the intensive care unit. The remaining patients (93.3%) either did not show any signs of bleeding on the postoperative CT or had only minor traces of hemorrhages, which did not concur with neurological deficits. In all the patients, no anaphylactic or other adverse events in association with fluorescein administration were detected. Mean preoperative creatinine was 0.99 mg/dL (±0.39), and postoperative creatinine was 0.94 mg/dL (±0.41). Hence, creatinine was not significantly altered after the i.v. application of fluorescein.

### 3.3. Patterns of Fluorescein Uptake and Tumor Pathology

In total, 63% of the acquired specimens were fluorescein positive. Of these fluorescein-positive samples, 82% were either tumor-positive (or yielded a pathological diagnosis if not of tumor-entity). Figure 2 shows the detailed fluorescein uptake pattern based on histology and gadolinium uptake. Samples were categorized in four subgroups: (1) fluorescein-positive and tumor/diagnosis-positive (F+T+), (2) fluorescein-positive and tumor/diagnosis-negative (F+T-), (3) fluorescein-negative and tumor/diagnosis-positive (F-T+), and (4) fluorescein-negative and tumor/diagnosis-negative (F-T-).

The fluorescein uptake based on final pathology was of a heterogenous nature (Figure 2A,B). In the glioblastoma tumor and the B-cell NHL group, almost three-quarters of the samples harvested were both fluorescein- and tumor-positive. The sensitivity (fluorescein-positive tissue is tumor- or diagnosis-specific) was 85% and 82% for GBM IV and for B-cell NHL, respectively, and the specificity (fluorescein-negative tissue is tumor-negative) was 70% and 44% for GBM IV and for B-cell NHL, respectively. Hence, the number needed to sample to achieve a definitive pathological diagnosis was 2 for GBM IV and 4 for B-cell NHL.

### 3.4. Correlation of Fluorescein Uptake Patterns with Gadolinium-Enhancement

Fluorescein uptake characteristics based on contrast-enhancement (gadolinium) on MRI is shown in Figure 2C,D. For contrast-enhancing lesions, the sensitivity of fluorescein was 84%, and the specificity was 50%. To obtain a histopathological diagnosis, the number needed to sample for contrast-enhancing lesions was three, whereas non-contrast lesions showed a sensitivity for fluorescein of 12% and a specificity of 72%. In the metastatic lesions, which all demonstrated contrast-enhancement, all samples were fluorescein- and tumor-positive.

As expected, in the AA III and the OD II group, representing lesions with heterogenous or no contrast-enhancement pattern, proportionately fewer samples were fluorescein-positive. The sensitivity in both of these entities was 33%. Only one of the six AA III cases showed contrast agent uptake in the MRI. Interestingly, in this subgroup, in which 57 biopsy samples were harvested, 33 (58%) demonstrated fluorescein uptake, and 82% of these fluorescein-positive samples were also tumor-positive. Taking only the Gadolinium-negative samples in this subgroup into account, 22/46 (48%) samples were fluorescein-positive, of which 73% were tumor-positive. The overall number needed to sample, whether contrast-enhancing or not, was five for final histopathological diagnosis.

### 3.5. Illustrative Case

A 76-year-old patient with a left-sided lesion in the corpus callosum underwent frameless biopsy using the VarioGuide^®^ system. The preoperative MRI showed a heterogenous gadolinium-enhancing lesion with perifocal edema in the splenium of the corpus callosum on T2 FLAIR-weighted images (Figure 3A,B). A total of eight biopsies were taken, two of which are exemplarily shown: sample 1 was taken from the FLAIR-positive peritumoral area representing focal edema and/or tumor infiltration zone, and sample 2 was obtained from the tumor core within necrosis. Figure 3D,E shows the intraoperative trajectory of the two samples. Under the Yellow560 filter of the Pentero 900 microscope, a bright yellow signal was seen in sample 2, which was taken from the tumor core, while sample 1 showed no fluorescein uptake (Figure 3C).

## 4. Discussion

In this study, we share our experience with the use of fluorescein in stereotactic biopsies and reveal its uptake characteristics in correlation to histopathological features. To our knowledge, this is the largest series reporting the utilization of fluorescein in stereotactic neurosurgical cases.

In contrast to previous studies on the use of sodium fluorescein in neurosurgical biopsies, we sampled multiple specimens per patient to explore the dye’s uptake patterns and its sensitivity, specificity, and potential predictive accuracy in pathological diagnosis with a total number of 347 samples analyzed. As expected, our data revealed a heterogenous fluorescein uptake pattern depending on final pathology and gadolinium-enhancement on MRI. These heterogenous patterns of fluorescein uptake are somewhat expected since our series also includes a vast range of different intracranial pathologies, not only limited to contrast-enhancing tumors. We have previously shown that the heterogenous fluorescein uptake pattern on a microscopic and spectroscopic level is influenced by the dye’s concentration, pH sensitivity, and variability in the tumor microenvironment [17]. These variations in uptake may also be influenced by additional factors that need to be taken into consideration, such as the routine use of intraoperative mannitol in neurosurgical procedures that may increase fluorescein uptake as it leads to alterations in the blood-brain barrier [18]. However, as an institutional standard, mannitol was not used in the stereotactic biopsies in our series.

Moreover, our data demonstrate a potential use of fluorescein in a range of primary and secondary brain tumors, as well as non-tumorous lesions, especially when these lesions show contrast-enhancement. The two entities that showed very similar uptake patterns where glioblastoma (WHO grade IV), as well as CNS lymphomas. This is somewhat expected since both lesions classically are contrast-enhancing on MRI. To have a clearer analysis that is helpful for an a priori setting where histology still remains unclear before biopsy, we also categorized our analysis into contrast-enhancing and non-enhancing lesions. In gadolinium-enhancing lesions, the sensitivity of fluorescein was 84% with a number 3 needed to sample for a robust diagnostic yield. The practical and pragmatic implication here is that if a contrast-enhancing lesion is biopsied, adding the modality of placing the sampled tissue under the Yellow560 filter and finding that three of these are fluorescein positive, the likelihood of a final, conclusive histopathological diagnosis is robust. Even in non-contrast-enhancing lesions, fluorescein harbors a specificity of 72%. Hence, our data correspond with previous studies that have also reported on usefulness of fluorescein in stereotactic biopsies by increasing the diagnostic yield and shortening operative time, although they mainly focused on contrast-enhancing lesions [11,14,19].

An interesting group in our study cohort comprised the astrocytoma, IDH-mutant WHO grade III group, in which only one out of six cases showed Gadolinium uptake on preoperative MRI—yet 22/46 (48%) demonstrated fluorescein uptake despite no evidence for gadolinium uptake, and 73% of these were in fact tumor-positive. As discussed in previous studies, the uptake of fluorescein in non-contrast lesions is possibly explained by its uptake biology and discrepancy in molecular weight compared to gadolinium: sodium fluorescein bears a molecular weight of 376 Da, whereas gadolinium has a larger molecular weight of 552 Da [13,20,21].

In this series, there were four cases in total in which the pathology revealed “reactive CNS tissue”. One of the cases was a repeat stereotactic biopsy in the same patient, which remained tumor-negative in the second biopsy. In all these cases, the underlying lesion showed Gadolinium enhancement, 25/33 specimens were fluorescein positive, with all 33 samples conclusive for the final histopathological diagnosis and having a sensitivity of 76%. In the further clinical course, the patients did not show signs of local tumor growth, conferring to the truly tumor-negative diagnoses.

The multiple sampling technique used in our study poses the question of increased risk for hemorrhagic events. In the literature, the frequency of hemorrhages following stereotactic biopsy on postoperative CT ranges between 0.7–11.7% (and even up to 54.9% in a prospective study), while hemorrhage-associated morbidity has been reported with an incidence of 0.7 to 6.3% [22,23,24,25,26]. Hence, our complication rate of 6.7% is comparable to previous reports, despite the multiple samplings used herein. Furthermore, our data confirm that the use of fluorescein in stereotactic biopsies does not harbor risks for additional adverse events on kidney function with no significant alterations in creatinine postoperatively [27].

Our current study is limited by the inherent difficulties in translational research in human tissue, especially in such a pilot study setting—although biopsy specimen and fluorescein uptake pattern evaluation were conducted prospectively, the analysis was performed in a retrospective fashion. Furthermore, the range of neoplastic and non-neoplastic entities in our study and variations in sampling techniques add to the heterogeneity of the data. Finally, although Fluorescein is FDA-approved and numerous studies have demonstrated its safety, its application in neurosurgery still remains off-label.

In summary, this study demonstrates that fluorescein-guided stereotactic biopsy increases the diagnostic yield with a reduction of the number of biopsies needed to sample by 50% in contrast-enhancing lesions, and this may even correlate with tumor pathology in low-grade gliomas without gadolinium uptake, underpinning its predictive value in conclusive and accurate histopathological diagnosis. Our study is, however, limited by its retrospective nature, the heterogeneity of the pathologies, and the variations in sampling techniques. Further studies are needed to fully understand the biology of uptake mechanisms, especially in CNS lesions without gadolinium enhancement.

## Figures and Tables

**Figure 1 jcm-11-03330-f001:**
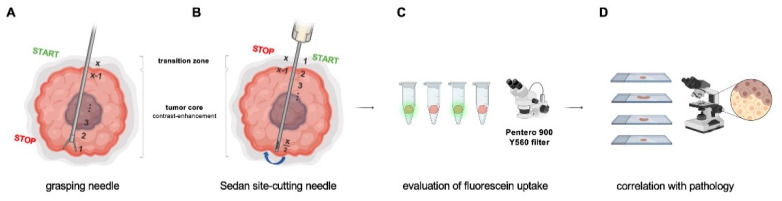
Biopsy technique and workflow. (**A**) Using a grasping needle of the Riechert–Mundinger system, the first biopsy was typically taken in the most distant part of the tumor, and for each consecutive specimen, the next trajectory was moved proximally through the tumor core until the transition zone. (**B**) When using the Sedan side-cutting needle, the initial biopsy was taken in the transition zone and then further advanced to the tumor core until the tumor margin. The needle was then turned 180° and samples were taken in a step-wise retrograde fashion. (**C**) Each tumor sample was evaluated under the Pentero 900 Y560 filter for fluorescein uptake. (**D**) Fluorescein uptake patterns were then correlated with histopathological diagnosis (*x* = number of total samples per patient).

**Figure 2 jcm-11-03330-f002:**
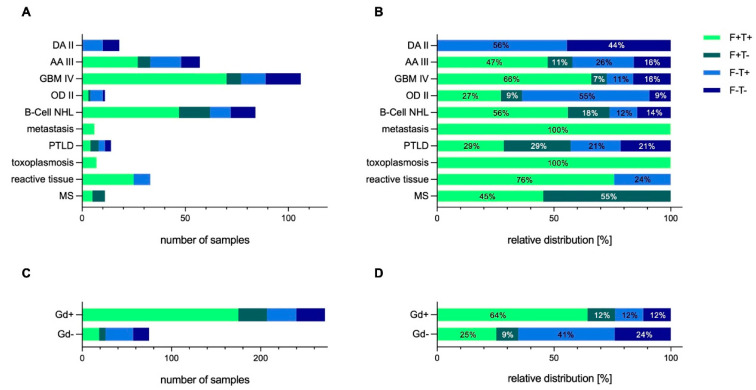
Fluorescein uptake patterns based on pathology and Gadolinium (Gd)-enhancement. (**A**,**B**) Distribution of Fluorescein uptake and diagnostic yield based on pathology. (**C**,**D**) Fluorescein uptake and diagnostic yield depending on Gadolinium uptake on MRI. Samples were categorized in four categories: (1) fluorescein-positive and tumor/tissue diagnosis positive (F+T+), (2) fluorescein-positive and tumor/tissue diagnosis negative (F+T-), (3) fluorescein-negative and tumor/tissue diagnosis positive (F-T+), (4) fluorescein-negative and tumor/tissue diagnosis negative (F-T-). Abbreviations: DA II = diffuse astrocytoma; IDH-mutant WHO grade II; AA III = anaplastic astrocytoma; IDH-mutant WHO grade III; B-Cell NHL = B-cell non-Hodgkin lymphoma; GBM IV = glioblastoma WHO grade IV; Gd = gadolinium; MS = multiple sclerosis; OD II = oligodendroglioma; IDH-mutant and 1p/19q-codeleted WHO grade II; PTLD = post transplantation lymphoproliferative disorder.

**Figure 3 jcm-11-03330-f003:**
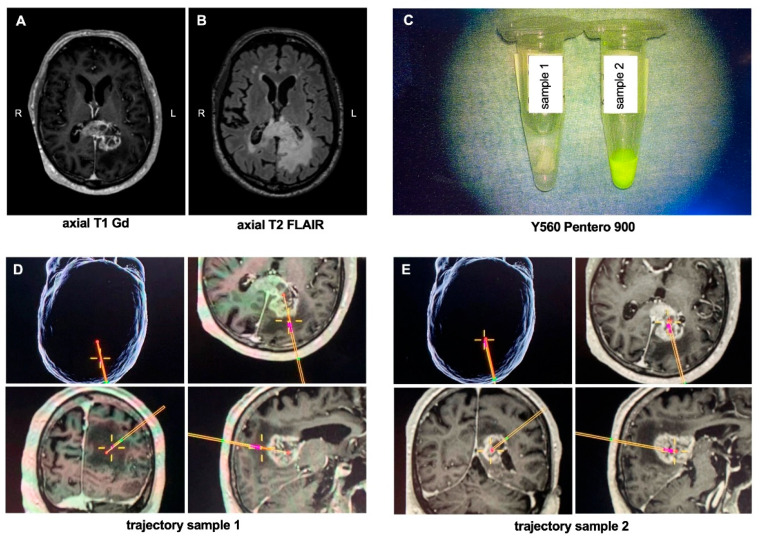
Illustrative case. (**A**) Axial Gadolinium-enhanced T1-weighted and (**B**) T2 FLAIR MRI of a 72-year-old patient who presented with a left-sided contrast-enhancing lesion with perifocal edema of the splenium of the corpus callosum. The patient underwent a VarioGuide^®^ biopsy in which a total of 8 samples were taken, of which two are exemplary shown: (**C**) image of the specimen under the Y560 filter of the Pentero 900 microscope; (**D**,**E**) intraoperative trajectory of the two obtained samples; sample 1 was taken from the infiltration zone of the tumor without contrast-enhancement, whereas sample 2 was obtained from tumor core. Abbreviations: Gd = gadolinium; L = left; R = right.

**Table 1 jcm-11-03330-t001:** Patient characteristics.

Patient Characteristics	
**Age**	
median (range)	63.8 (18–87)
**Gender**	
Female	15
Male	29
**Pathology**	
oligodendroglioma, IDH-mutant and 1p/19q-codeleted (WHO grade II)	2
diffuse astrocytoma, IDH-mutant (WHO grade II)	1
anaplastic astrocytoma, IDH-mutant (WHO grade III)	6
glioblastoma, IDH-mutant and IDH-wildtype (WHO grade IV)	16
ependymoma	1
lymphoma	10
multiple sclerosis	1
metastasis	1
infection (toxoplasmosis)	1
PTLD	2
reactive tissue/inflammation	4
**Biopsy technique**	
frame-based	25
frameless	16
**Sample Number (ntotal = 383)**	
median (range)	8 (4–18)
**Creatinine** (average ± SD)	
preoperative	0.99 ± 0.39 mg/dL
postoperative	0.94 ± 0.41 mg/dL

## Data Availability

All original data of the study will be made available upon request.

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
