# Peer review of "Correlation of Tumor Pathology with Fluorescein Uptake and MRI Contrast-Enhancement in Stereotactic Biopsies"

_jcm, 2022, doi:10.3390/jcm11123330_

Round 1
Reviewer 1 Report
The manuscript entitled." Correlation of Tumor Pathology with Fluorescein Uptake and MRI Contrast-Enhancement in Stereotactic Biopsies " focused on the evaluation of a novel approach based on MRI contrast in stereotassic biopsies is well written and requires minor comments to be accepted for the publication:
- In the results section, could the authors define of a grading in fluorescent uptake was observed according to illnes severity?
- In the results section, specificity is quite low, please, could the authors discuss this data? could it represents an issue for clinical application?Could the authors provide solutions to improve this data?
- Could the authors discuss the limitations of this study? In my opinion, this may be considered a pilot study and should be tailored according to this point
Author Response
We would like to thank the academic editor for the the constructive comments. We have addressed all the concerns and comments. Below are our answers to the specific comments.
In the results section, could the authors define of a grading in fluorescent uptake was observed according to illnes severity?
In intracranial tumors, the illness severity depends heavily on the size and location of the tumor as well as tumor type, causing a variety of focal and generalized neurological symptoms [1]. Since fluorescein uptake itself is dependent of the presence of Gadolinium uptake, and while these Gadolinium-positive tumors can vastly range in size, location and perifocal edema, and such characteristics define illness severity, the grading itself of fluorescent uptake cannot be correlated with illness severity. However, we have strengthened this point in the discussion section.
In the results section, specificity is quite low, please, could the authors discuss this data? could it represents an issue for clinical application?Could the authors provide solutions to improve this data?
The specificity of fluorescein in tumorous lesions ranges in biopsies has been reported in the literature between 65% and up to 100% in contrast-enhancing lesions. Our series reports a sensitivity of 84% in contrast-enhancing lesions, which confers to the current literature. The sensitivity for low-grade lesions in our series was lower with 72% which is somewhat expected since these low-grade tumors usually do not enhance gadolinium on MRI imaging, and fluorescein uptake represents the result of extracellular accumulation in a damaged blood-brain barrier (which is often not the case for low-grade lesions) [2]. In our view, this range of sensitivity does not hinder its clinical application but should be considered when interpreting fluorescence uptake in biopsied specimen.
Could the authors discuss the limitations of this study? In my opinion, this may be considered a pilot study and should be tailored according to this point
We have added this paragraph in the discussion section to elaborate more on the limitations of the study:
“Our current study is limited the inherent difficulties in translational research in human tissue, especially in such pilot study setting – although biopsy specimen and fluorescein uptake pattern evaluation was conducted prospectively, the analysis was done in a retrospective fashion. Furthermore, the range of neoplastic and non-neoplastic entities in our study and variations in sampling technique adds to the heterogeneity of the data. Finally, although Fluorescein is FDA approved and numerous studies have enlightened its safety, its application in neurosurgery still remains off-label.”
We thank the reviewer for the thoughtful comments that helped to elevated the quality of the manuscript.
Reviewer 2 Report
First of all thanks to the authors for their efforts to write this paper. This is nice work regarding fluorescein uptake in stereotactic biopsies. The study has a good structure and it´s well written. The main positive aspects of this paper are related to its contribution to the search for improvements in biopsy techniques for the benefit of the best diagnosis with less morbidity.
Nevertheless, I see some minor problems like:
- The heterogeneity of the pathological diagnoses. These are very different entities under the microscope and this may be a problem to find a really significant relation between the uptake and the diagnosis.
- And maybe the evaluation 'per se' of the fluorescein uptake. How this was reported or evaluated? there was a scale of positivity? something like positive 1/3 or just positive or negative?
Even if this is a retrospective work (as many are and this does not represent a significant problem if the details are taken into account), I think these two observations are the ones that kept me thinking.
My recommendation regarding the first one is to take into account if there is concordance with some histopathological pattern with that uptake. What do Lymphomas and WHO-grade IV tumors have in common? maybe the key to finding more specificity is to be aware of the kind of histopathological pattern and the fluorescein uptake, even if the sampling technique is different.
Regarding the second observation, a possible solution may be to explain a bit better how the positivity was evaluated and reported intraoperatively so the findings on the uptake can be more reproducible.
Thanks again for your effort.
Author Response
We would like to thank Reviewer #2 for their thoughtful comments and kind consideration of our manuscript. We have addressed the two main points in the following:
We agree with the important point made here by the reviewer. This study, indeed, includes a heterogenous range of different entities. Thus, we also group for the most relevant ‘a priori’ factor in the clinical setting, namely contrast-enhancement and non-Gadolinium enhancement of the tumors in Fig. 2C and 2D. The uptake patterns in the different pathologies as shown in Fig. 2A and 2B is an effort to depict a quick graphical overview of the uptake characteristics in each entity but we do agree, that it should not be over-analyzed. The second point made by the reviewer is of relevance – lymphomas and WHO grad IV tumors are classically Gadolinium-enhancing lesions and since they present similar contrast-enhancing patterns on MRI, it is somewhat expected that their fluorescein uptake patterns (which correlates also with blood brain barrier breakdown) is similar. We have added some additional information on that in the discussion section:
„ Moreover, our data demonstrate a potential use of fluorescein in a range of primary and secondary brain tumors, as well as non-tumorous lesion, especially when these lesions show contrast enhancement. The two entities that showed very similar uptake patterns where glioblastoma (WHO grade IV) as well as CNS lymphomas. This is somewhat expected since both lesions classically are contrast-enhancing on MRI. To have a clearer analysis that is helpful for an a-priori setting where histology still remains unclear before biopsy, we also categorized our analysis into contrast-enhancing and non-enhancing lesions. In gadolinium-enhancing lesions, the sensitivity of fluorescein was 84% with a number needed to sample of 3 for a robust diagnostic yield. The practical and pragmatic implication here is that if a contrast-enhancing lesion is biopsied, adding the modality of placing the sampled tissue under the Yellow560 filter and three of these are fluorescein positive, the likelihood of a final, conclusive histopathological diagnosis is robust.“
In regard to the evaluation of the fluorescein uptake itself, this is also an important note by the reviewer. Fluorescein positivity was classified by the surgeon into a) “no uptake”, b) “weak uptake”, and c) “strong uptake”. Our surgeries are typically carried out by a team of two neurosurgeons (resident+attending) who evaluated for fluorescein uptake and we found that there was essentially always agreement upon uptake of fluorescein. For the final statistical analysis, we then categorized into a binary classification (0 = “no uptake”; 1 = “uptake”). This decision was also based on a recent study that has shown that the observers agreed on 95% of thebiopsy samples in terms of their fluorescing status, denoting a substantial inter- observer agreement (κ = 0.77) [3].
Finally, we thank the reviewer for the thoughtful comments that helped us to revise the manuscript, elevating the clarity and quality.
Literature
1. DeAngelis, L.M. Brain tumors. N Engl J Med 2001, 344, 114-123, doi:10.1056/NEJM200101113440207.
2. Diaz, R.J.; Dios, R.R.; Hattab, E.M.; Burrell, K.; Rakopoulos, P.; Sabha, N.; Hawkins, C.; Zadeh, G.; Rutka, J.T.; Cohen-Gadol, A.A. Study of the biodistribution of fluorescein in glioma-infiltrated mouse brain and histopathological correlation of intraoperative findings in high-grade gliomas resected under fluorescein fluorescence guidance. J Neurosurg 2015, 122, 1360-1369, doi:10.3171/2015.2.JNS132507.
3. Nevzati, E.; Chatain, G.P.; Hoffman, J.; Kleinschmidt-DeMasters, B.K.; Lillehei, K.O.; Ormond, D.R. Reliability of fluorescein-assisted stereotactic brain biopsies in predicting conclusive tissue diagnosis. Acta Neurochir (Wien) 2020, 162, 1941-1947, doi:10.1007/s00701-020-04318-5.